# Theater in the Self-Cleaning Cell: Intrinsically Disordered Proteins or Protein Regions Acting with Membranes in Autophagy

**DOI:** 10.3390/membranes12050457

**Published:** 2022-04-24

**Authors:** Hana Popelka, Vladimir N. Uversky

**Affiliations:** 1Life Sciences Institute, University of Michigan, Ann Arbor, MI 48109, USA; 2Department of Molecular Medicine, Byrd Alzheimer’s Research Institute, Morsani College of Medicine, University of South Florida, Tampa, FL 33612, USA; vuversky@usf.edu

**Keywords:** amphipathic helix, disorder prediction, hydrophobic finger, intrinsically disordered protein, intrinsically disordered protein region, lipid bilayer, membrane binding, membrane remodeling, membrane tethering, membrane fragmentation, protein–protein interactions

## Abstract

Intrinsically disordered proteins and protein regions (IDPs/IDPRs) are mainly involved in signaling pathways, where fast regulation, temporal interactions, promiscuous interactions, and assemblies of structurally diverse components including membranes are essential. The autophagy pathway builds, de novo, a membrane organelle, the autophagosome, using carefully orchestrated interactions between proteins and lipid bilayers. Here, we discuss molecular mechanisms related to the protein disorder-based interactions of the autophagy machinery with membranes. We describe not only membrane binding phenomenon, but also examples of membrane remodeling processes including membrane tethering, bending, curvature sensing, and/or fragmentation of membrane organelles such as the endoplasmic reticulum, which is an important membrane source as well as cargo for autophagy. Summary of the current state of knowledge presented here will hopefully inspire new studies. A profound understanding of the autophagic protein–membrane interface is essential for advancements in therapeutic interventions against major human diseases, in which autophagy is involved including neurodegeneration, cancer as well as cardiovascular, metabolic, infectious, musculoskeletal, and other disorders.

## 1. Introduction

Research in recent decades has revealed that a large fraction of proteins or protein regions does not form a well-defined three-dimensional structure, but is still functional. These polypeptides are referred to as intrinsically disordered proteins (IDPs) or hybrid proteins containing ordered domains and intrinsically disordered protein regions (IDPRs) [1,2,3,4]. Many studies and reviews on IDPs and IDPRs have uncovered their amazing physiochemical properties and indispensable cellular functions as well as a large pharmacological potential [4,5,6,7,8,9,10,11,12]. This new knowledge makes us realize how little we knew about the eukaryotic proteome before IDPs and IDPRs were recognized. We are at the beginning of disclosing the complexity of the interplay between lipid bilayers and protein disorder [13]. Membrane properties such as lipid composition, curvature, or cellular localization affect the recruitment of IDPs/IDPRs and, vice versa, disordered polypeptides determine which membrane will be tethered, curved, pinched off, or fragmented.

In this review, we focused on the interface between IDPs/IDPRs and membranes in autophagy, a highly conserved lysosomal degradation pathway that maintains cellular homeostasis. The pathway mediates the clearance of cellular materials such as protein aggregates, intracellular pathogens, and damaged or excessive organelles [14,15], and protects humans from diseases [16]. During the autophagy process (Figure 1), a double-membrane organelle, called the autophagosome, is formed de novo around the cargo that is destined for degradation. Nonspecific cargo is degraded by a bulky process, macroautophagy, whereas specific cargo is cleared by selective types of autophagy. For example, excessive or damaged mitochondria is cleared by mitophagy, the endoplasmic reticulum is degraded by reticulophagy, and protein aggregates are specifically removed by aggrephagy. The autophagy process is induced by a protein assembly called the Atg1/ULK1 initiation complex that recruits initial factors and membranes. This step leads to the formation of the cup-shaped phagophore, which is an autophagosomal precursor. Recruitment of additional components of the autophagy machinery expands the phagophore in a phosphoinositide-dependent manner until the phagophore engulfs the cargo and becomes the autophagosome. The phagophore expansion requires conjugation of the essential protein, Atg8 (LC3 or GABARAP in higher eukaryotes), to the primary amino head group of phosphatidylethanolamine (PE) on the phagophore. Closure of the phagophore creates the autophagosome and a release of some autophagy components from the autophagosomal outer membrane allows for fusion of the mature autophagosome with the lysosome (vacuole in yeast), forming the autolysosome, an organelle where degradation occurs. The degradative products are released to the cytoplasm for cellular reuse (Figure 1).

Autophagy as a signaling pathway relies heavily on the multi-functionality of IDPs and IDPRs [17]. In the sections below, we describe the individual components of the autophagy machinery that come into various contacts with the membranes. We focused on detailed molecular mechanisms of the IDPs/IDPRs–membrane interplay and their manifestation in the autophagy pathway (Table 1).

## 2. Extended Disordered Conformations Bind to Membranes via Coupling of Electrostatic and Hydrophobic Interactions

An intrinsically disordered conformation of proteins or protein regions arises from a unique amino acid composition that yields a high mean net charge and low mean hydrophobicity [3], which in turn grants the IDP sequences their important functionalities [4,7,8,9,10,11,12,18]. When protein domains, often loops, in extended (unfolded, lacking a transient secondary structure) disordered conformations interact with anionic membranes in the course of autophagy, they do so via a combined effort of positively charged amino acid residues and bulky hydrophobic side chains that operate as aromatic fingers, which insert deep into the hydrophobic portion of the lipid bilayer. A typical example of this mechanism exists in Atg6/BECN1, an essential subunit of the PtdIns3K-CI complex in yeast and higher eukaryotes. The C-terminus of Atg6/BECN1 folds into a globular BARA domain (residues 245–450 in *H. sapiens* or residues 320–539 in *S. cerevisiae*). The BECN1 BARA includes a disordered loop with an aromatic triad, _359_FFW (human numbering), forming a hydrophobic finger that inserts into the lipid bilayer and mediates protein binding to liposomes. The triad is immediately preceded by arginine and followed by the _364_KFDH sequence at the beginning of the α3 helix [19]. A similar function was found in the Atg6(Vps30) BARA domain from yeast (Figure 2A,C,E). Experiments showed that a segment (residues 424–443) in this domain, which involves a loop and a portion of the α3 helix including the sequence _428_RIFRKETKFDK (*S. cerevisiae* numbering) conserved in yeast species, exhibited a decreased hydrogen-deuterium exchange rate in the presence of liposomes [20] (Figure 2E), suggesting that the membrane masks this segment. This means that the BARA in Atg6(Vps30) and BECN1 utilizes a similar membrane-binding region [19,20] and that the mechanism is evolutionarily conserved from yeast to human.

Another globular domain that utilizes an intrinsically disordered loop for membrane anchoring is a Phox homology (PX) domain. Autophagy related proteins such as Vam7, Atg20, or Snx4(Atg24) use PX domains for selective recruitment to phosphatidylinositol 3-phosphate (PtdIns3P)-enriched membranes because these domains have extensive, positively charged binding pockets for specific PtdIns3P ligation [32]. In Vam7, the PtdIns3P ligation by the PX domain is dominated by electrostatic interactions, however, these interactions are further enhanced by an extended loop that connects the α1 and α2 helices of the PX domain [33]. The loop carries a proline-rich motif, PxxP, which is also conserved in Atg20 and Snx4(Atg24). This motif likely facilitates the extended conformation of the loop, as proline is the most intrinsic disorder-promoting residue [34]. We call this loop a PX-GAP. In the Vam7 PX domain, a cooperation of polar and hydrophobic residues in the PX-GAP, positioned between the PxxP motif and the α2 helix, forms a direct membrane binding element that stably anchors the PX domain to the lipid bilayer. An alignment of amino acid sequences from Vam7, Atg20, and Snx4(Atg24) revealed that Snx4(Atg24) has a homologous PX-GAP of a similar length as that in Vam7, but Atg20 has a PX-GAP that is much longer [32] (Figure 3A). It is plausible to assume that the long Atg20 PX-GAP adopts some additional membrane-inducible structures to allow the phosphoinositide-binding pockets in the PX to reach PtdIns3P on the membrane. How exactly the Atg20 PX-GAP is packed at the membrane interface remains to be determined.

A high net charge in IDPRs [3] is a useful tool for protein association with membranes when positively charged amino acid residues are robustly combined with the phenylalanine side chains in order to bind to negatively charged phospholipids. Grouping of lysines and phenylalanines into a short linear motif in an extended conformation was found to be an efficient membrane anchor in the large intrinsically disordered region of Atg13 from yeast (Figure 4A,C). The interaction of this polypeptide with phospholipid membranes is mediated by the two motifs, _640_KFK and _683_KFHK, where lysines employ electrostatic forces and bulky phenylalanine residues insert into the hydrophobic membrane core [35] (Figure 4E). This mechanism is analogous to that found in the myristoylated alanine rich protein kinase C substrate (MARCKS) [36]. For Atg13, mutagenesis that puts glutamic acid or glycine in the membrane-binding motifs yields substantially reduced binding to liposomes and defects in autophagy flux [35]. Interestingly, the membrane binding segment in Atg13 IDPR overlaps with the region that binds to Vac8, a multifunctional vacuolar protein. However, the Vac8-binding domain in Atg13 is mediated by a different set of amino acid residues and the interactions of Atg13 with lipids or Vac8 are mutually exclusive. Cationic residues in the Vac8 groove bind to the Atg13 IDPR constitutively under growing conditions. A mechanism through which Atg13 is released from Vac8 in order to bind to the membrane has not been elucidated. However, it is known that upon starvation conditions, the Atg13 IDPR is dephosphorylated at multiple sites including the region between the residues 640–660. Therefore, a possible mechanism that may facilitate the release of Atg13 from Vac8 and enhance the interaction of the lipid-binding motifs, _640_KFK and _683_KFHK, with the membrane is dephosphorylation. It seems plausible that a purpose for the overlap of the lipid- and Vac8-binding regions in Atg13 can be to function as a fast mechanistic switch (Figure 4E,F). It should be noted that such a switch has to be carefully regulated because the Vac8–Atg13 interaction is important for retaining the liquid phagophore assembly site (PAS) near the vacuole. Specifically, a recent study showed that the Atg1 complex, enriched in IDPRs, induces liquid–liquid phase separation and that the Atg1-complex droplets are tethered to the vacuolar membrane via the Vac8–Atg13 interaction [37].

## 3. Disorder-To-Order Transitions at the Protein-Membrane Interface

Lipid bilayers can significantly modulate the structural flexibility of disordered regions in proteins. Studies probing mechanisms of interactions at the protein–membrane interface show that specific amino acid sequences in numerous IDPs or IDPRs fold into a transient secondary structure, typically an amphipathic α-helix (AH). Such a helix buries its hydrophobic face into the lipid bilayer, while the polar, hydrophilic, face remains exposed above the membrane surface. If a polar residue is lysine, a positive amine group tends to interact with negatively charged polar head groups of anionic membranes, whereas a long hydrophobic side chain can contribute to the interaction with the hydrophobic core of the lipid bilayer [38]. This behavior of the lysine side chain is known as the snorkel phenomenon. If the polar face of an AH is enriched in glutamate residues, an anionic membrane that attracts protons can cause glutamate protonation, which increases hydrophobicity of the nonpolar face, and thereby, enhances the binding affinity of an AH to the membrane [39,40].

One of the first proteins in the autophagy pathway discovered to utilize an amphipathic α-helix in membrane binding is ATG14, a subunit of the mammalian VPS34-containing class III phosphatidylinositol 3-kinase complex I (PtdIns3K-CI). The VPS34 complex generates PtdIns3P, a molecule that is required on autophagic membranes for recruitment of downstream effectors of the autophagy pathway. The disordered C-terminal tail of ATG14 (Figure 2B,D,E) carries an autophagosome targeting sequence called BATS (residues 413–492 in *S. cerevisiae*), which is predicted to be mostly disordered in the unbound state (see Figure 2B). Upon contact with the phagophore membrane, a portion of BATS folds into an amphipathic helix spanning the region between Gly471 and Y488 [41]. The hydrophobic face in the AH relies on a set of three aromatic amino acid residues, W484, F485, and Y488, which are critical for the association of ATG14 with the phagophore (Figure 2B,D,E). Experiments with GFP-tagged BATS in cells showed that this domain has an ability to sense and stabilize membrane curvature. However, the BATS AH is not the only element that keeps the protein on the phagophore. After recruitment to the membrane, a coiled-coil domain of Atg14/ATG14 (residues 71–180 in *H. sapiens*, predicted to be mostly disordered, see Figure 2B) engages with a coiled-coil domain of Atg6/BECN1 (residues 142–279, predicted to be mostly disordered, see Figure 2A) to form the stable Atg6-Atg14/BECN1-ATG14 heterodimer on the phagophore. Therefore, this heterodimer integrates the two IDPR-based membrane-anchoring mechanisms, the aromatic finger in the Atg6/BECN1 BARA and the amphipathic α-helix at the Atg14/ATG14 C-terminus (Figure 2E).

Another amphipathic element that acts in an early stage of autophagy is found in Atg20, a subunit of the Atg1 initiation complex in yeast, which consists of the Atg1 serine/threonine kinase, Atg13, the Atg17-Atg29-Atg31 dimeric trimer, and supporting subunits such as Atg11, Vac8, and the Atg20-Snx4(Atg24) heterodimer. The two polypeptides in the heterodimer are sortin nexins, containing a PX and Bin/Amphiphysin/Rvs (BAR) domain, where Snx4(Atg24) is predominantly a globular protein. In contrast, Atg20 contains a significant amount of intrinsic disorder, namely the N-terminus, the PX-GAP in the PX domain, the flexible PX-BAR linker, and a loop in the BAR domain termed the BAR-GAP (Figure 3A). The BAR-GAP (residues 487–574, *S. cerevisiae* numbering) contains an amino acid sequence that folds into an AH upon contact with the membrane (Figure 3A-C). In the Atg20 AH, F539 and F542 are instrumental for burying the hydrophobic face of the AH into the lipid bilayer [42]. In general, PX-BAR sortin nexins remodel membranes, and amphipathic helices are important for membrane tubulation (which is a membrane organization process leading to the formation of a tubular projection) [43]. Negative-stain electron microscopy images showed that the Atg20-Snx4 heterodimer tubulates membranes, and that the double mutation in the AH (F539E F542E) of Atg20 renders significantly lower tubulation efficiency compared to the membrane remodeling observed for the wild type of this protein. The position of the membrane-inducible AH in the Atg20 BAR-GAP is as-yet unknown in sortin nexins, and it remains to be determined whether it is autophagy-specific. In any case, Atg20 appears to be a unique PX-BAR protein, where the two globular domains, the PX and BAR, are interrupted by disordered regions that attach the protein to lipid bilayers (Figure 3C).

The formation of vesicles in the cytoplasm-to-vacuole targeting (Cvt) pathway, a biosynthetic pathway in yeast that involves the autophagy machinery as well as the formation of autophagosomes in selective and nonselective autophagy, requires a supply of lipids. Two proteins, Atg2 and Atg18, in yeast (ATG2A-WIPI4 in mammals), are part of the Atg9/ATG9 complex (Figure 1), which plays an important role in supplying lipids to the growing phagophore [44,45,46]. The Atg2–Atg18/ATG2A–WIPI4 lipid-transporting complex is recruited to the phagophore assembly site in a PtdIns3P-dependent manner, and functions as a membrane tether at the junction of the phagophore and the ER membrane [44,45,46]. Atg2/ATG2A anchors the Atg2–Atg18/ATG2A–WIPI4 complex to the ER via an N-terminal tip (Figure 5A,C,E) that folds into a lipid-transfer-protein-like hydrophobic cavity capable of accommodating phospholipid acyl chains. In contrast, phagophore-targeting sequences in Atg2 and Atg18 arise from disordered regions, and each forms an amphipathic α-helix. Specifically, the disordered C-terminal segment in Atg2 from yeast (residues 1347–1373) acts as a phagophore anchor positioned in the large Atg2 molecule opposite the N-terminal tip. Therefore, Atg2 acts as a β-sheet “tunnel” that transfers lipids from one type of the membrane, the ER, to the other, the phagophore (Figure 5E). In comparison, Atg18 is a seven-bladed β-propeller that carries an intrinsically disordered loop in the sixth blade [47]. This IDPR folds into an amphipathic α-helix by insertion into the lipid bilayer of the phagophore [48] (Figure 5B,D,E). A conformational transition of the disordered loop into the AH has several important functional aspects. First, the Atg18 AH is important for vacuolar scission. Second, the AH acts as a negative regulator of membrane binding because deletion of the disordered loop leads to excessive binding of Atg18 to the membrane. This observation is consistent with the presence of serine residues in the hydrophobic face of the AH and the known phosphorylation sites in the region. Third, the AH in Atg18 appears to be a sensor that contributes to differentiation between PtdIns(3,5)P2 and PtdIns3P. This function comes from the fact that blades 5 and 6 in the Atg18 β-propeller create binding pockets that attach to PtdIns(3,5)P2 or PtdIns3P on the membrane, yielding phosphoinositide-dependent recruitment of the Atg2–Atg18 complex. Finally, Atg18 AH has been proposed to eliminate steric interference of the disordered mass, thereby, facilitating engagement of the two phosphoinositide-binding sites in Atg18 [48].

Atg18 (WIPI4 in mammals) has an orthologue, Atg21 (WIPI2 in mammals) (Figure 6A,C,E). Membrane binding of the Atg21/WIPI2 β-propeller is membrane curvature dependent [47] as the protein utilizes an AH in concert with phosphoinositide-binding sites [48,49,50,51]. Atg21 was found at the highly curved edge of the phagophore and the vacuole, called the vacuole-isolation membrane contact site [52]. Because PtdIns3P-binding pockets in Atg21 are insensitive to membrane curvature, recruitment of Atg21 at the highly curved vacuole-isolation membrane contact site is likely mediated by the Atg21 AH.

Atg21/WIPI2 recruits Atg16/ATG16L1 to the phagophore [52,53]. Atg16 in yeast (Figure 6B,D,E) is a simple intrinsically disordered protein that undergoes disorder-to-order transitions in three separate segments of its amino acid sequence, the N-terminal region in order to bind Atg5, the middle segment to dimerize via the coiled-coil formation and the C-terminal segment to bind autophagic membranes. The latter segment folds into an amphipathic α-helix that makes Atg16 exclusively membrane associated. A mutagenic interference with the hydrophobic face of the AH completely abolishes nitrogen starvation-induced autophagy, and renders the cytosolic protein susceptible to degradation [54]. Mammalian ATG16L1 (Figure 7A,C,E) is a larger and more complex protein than its yeast homolog, but the amphipathic α-helix in ATG16L1 folded from the intrinsically disordered N-terminus is also indispensable for functional autophagy [55]. Interestingly, crystallographic studies [56,57] revealed that the ATG16L1 AH is immediately downstream of the ATG5-binding region, and, in the absence of membranes, conceals its hydrophobic face either in the ATG5–ATG16L1 interface or in the crystal-induced ATG16L1 dimer. This observation not only shows the binding promiscuity of the IDPR spanning the AH sequence, but also underscores the importance of studying folding-prone disordered regions in the presence of their physiological binding partner, which, in the case of the ATG16L1 AH, is the membrane (Figure 7E).

The proteins described in previous paragraphs comprise the core autophagy machinery that orchestrates autophagosome biogenesis as well as engulfment and clearance of the cargo in the enclosed autophagosome. However, there are many cellular proteins that interact with the core autophagy machinery and function as autophagy adaptors, receptors, or regulators. Proteins that facilitate degradation of specific types of cargo via selective autophagy are called autophagy receptors. As such, they connect the cargo with Atg8 proteins conjugated to PE on the phagophore [58,59]. If the cargo is a membrane, the receptor is a membrane-embedded protein. For example, degradation of the endoplasmic reticulum (ER) requires reticulophagy receptors inserted into the ER. One of these receptors in mammalian cells is RETREG1. It consists of the reticulon-homology domain (residues 84–233 in *H. sapiens*) flanked at the N- and C-termini by intrinsically disordered regions (residues 1–63, 160–400, and 431–497, which account for >55% of the amino acid sequence of this protein, see Figure 8A,C). The reticulon-homology domain is dynamic, and requires the presence of membranes to adopt its final conformation [60,61]. Molecular dynamics simulations showed that four transmembrane helices together with two amphipathic α-helices of the RETREG1 reticulon-homology domain formed a wedge-shaped structure. Clustering of the reticulon-homology domain wedges strongly curved the ER. Such membrane remodeling is essential for fragmentation and ultimately engulfment of the organelle by the phagophore. The two amphipathic α-helices (residues 165–185 and 237–254) induced by the membrane in the IDPRs of the RETREG1 are inserted in the ER bilayer on the cytosolic side (Figure 8E).

Another autophagy receptor containing an AH for anchoring to membranes is NBR1. This highly disordered receptor (its overall disorder content is exceeding 60%, see Figure 8B,D) is involved in targeting ubiquitinated proteins for autophagosomal degradation. The NBR1 AH (residues 907–924) recruits the protein into the late endosomes. These vesicular structures fuse and ultimately become a part of the autophagosome, where the ubiquitinated cargo is degraded [62]. The NBR1 AH immediately precedes the C-terminal ubiquitin-associated domain (UBA) domain (Figure 8F).

Taken together, the examples of proteins described so far in this section illustrate that membrane-induced disorder-to-order transitions of IDPRs to amphipathic α-helices not only anchor proteins to their prospective lipid bilayer, but also cause an array of important membrane remodeling activities. Sensing or inducing membrane curvature by an AH is essential for membrane tubulation, fragmentation, organelle scission, or vesicle tethering. These processes occur in autophagy with membranes from various cellular sources such as the endoplasmic reticulum, Golgi apparatus, plasma membrane, cytoplasm-to-vacuole targeting (Cvt) vesicles (which are double membrane-layered vesicles implicated in the cytoplasm-to-vacuole targeting pathway), or endosomes. Tubulation and tethering ultimately lead to the formation of the autophagosome that engulfs various cargo for degradation, whereas scission and fragmentation facilitate a breakdown of large membrane organelles.

Next to binding and membrane remodeling, amphipathic helices arising from IDPRs can directly affect protein activity. This phenomenon was recently discovered in the E2-like conjugating enzyme Atg3/ATG3, which possesses several functionally important intrinsically disordered regions [63,64]. The very N-terminus of this protein is foldable into an AH (residues 3–16 in *H. sapiens*; Figure 7B,D,E) that targets the enzyme to the phagophore in a curvature-dependent manner [65,66]. A conserved disordered region, immediately downstream of the AH, of ten amino acids with a consensus motif, EYLTP, senses the membrane binding activity of Atg3/ATG3 and couples it with its conjugation activity. In other words, the EYLTP motif somehow communicates structural changes at the N-terminus upon membrane binding to rearrangements of the enzyme active site carrying the catalytic cysteine. A detailed structural basis of this event remains to be elucidated, and needs to be solved in the context of another allosteric activation of the Atg3/ATG3 enzyme, the one that involves repositioning of the catalytic cysteine allosterically communicated by the interaction between a large disordered loop of Atg3/ATG3 (Figure 7B) and the N-terminus of Atg7/ATG7 [64].

## 4. Membrane Anchoring of Disordered Proteins and Regions by Posttranslational Modifications

One of the advantages of protein intrinsic disorder is an easy access of posttranslational modification (PTM) enzymes to a disordered polypeptide chain [8,67,68,69,70,71,72,73]. There are multiple ways as to how posttranslational modifications facilitate or mediate association of an IDP/IDPR with the membrane. We mentioned above that phospholipid binding of Atg13 is compatible with protein dephosphorylation [35], although direct evidence still needs to be acquired. Some proteins of the autophagy machinery undergo a specific lipid modification to gain a fatty-acid membrane anchor. A typical example from yeast is Vac8, a peripheral membrane-bound protein that consists of 12 armadillo repeats downstream of the N-terminal disordered region of about 40 amino acid residues (Figure 4B,D). The N-terminal IDPR partially folds into the α1-helix, the position of which depends on the binding partner. The α1-helix ranging residues 20–41 packs at the first armadillo repeat when Vac8 interacts with Atg13, but folds at positions 19–35 when Vac8 binds to Nvj1, an interactor for the formation of the nucleus–vacuole junction [74]. The disordered segment of residues 1–18 is highly accessible and includes Gly2, which is myristoylated, and Cys4, Cys5, and Cys7 (Figure 4F), which are palmitoylated. All four modifications are essential for anchoring Vac8 to the vacuolar membrane [75]. Alanine mutations in these residues yield a protein that cannot associate with the vacuolar membrane and fails in inducing the efficient Cvt and autophagy pathways [75]. These observations are in line with known correlations between the intrinsic disorder status of the region undergoing PTM and the expected palmitoylation efficiency [76].

Another type of posttranslational modification for membrane targeting is prenylation, the covalent attachment of a lipid to the free thiol of a cysteine side chain. In the autophagy pathway, prenylation takes place in RAB33B, an interactor of ATG16L1. RAB33B has the disordered C-terminal tail that is not involved in the interaction with ATG16L1, but carries Cys227 and Cys229, both of which are prenylated. This modification anchors the protein on the phagophore, and facilitates the recruitment of ATG16L1 [77].

Palmitoylation and prenylation are also modifications at the C-terminal Cys194 and Cys195, respectively, in human YKT6, a SNARE protein involved in autophagosome-lysosome fusion. The mutation of these residues to serine blocks autophagy flux, but does not prevent localization of YKT6 on the autophagosomes, because this localization is permitted by the Longin domain at the N-terminus of YKT6 [78]. Palmitoylation and prenylation of YKT6 mediate efficient autophagy flux because they are required for autophagosome-lysosome fusion.

## 5. Membrane Remodeling Orchestrated by Extended Disordered Conformations

IDPRs acting in autophagy can remodel membranes without a direct insertion into the lipid bilayer. Membrane bending, curving, and ultimately fragmentation can be triggered in protein assemblies specifically accommodated by an IDPR. Examples of this mechanism can be found in reticulophagy, autophagic degradation of the endoplasmic reticulum, where the ER membrane requires remodeling and fragmentation [60,79]. In yeast, this mechanism is utilized by Atg40, a dimerizing reticulophagy receptor interacting with Atg8 and consisting of a reticulon-like domain embedded in the ER and a disordered region at the C-terminus. Figure 9A,C and Figure 9B,D show the results of the intrinsic disorder status analysis of Atg8 and Atg40 proteins from yeast, respectively. Although Atg40 is predicted to be mostly ordered, it contains a long C-terminal IDPR and several flexible loops (see Figure 9B). The Atg8 protein shows the opposite behavior, containing a long IDPR at its N-terminus, followed by a globular ubiquitin-like (UBL) domain and the disordered C –terminal tail (see Figure 9A). The Atg40 IDPR binds to Atg8, which is conjugated to PE in the dimeric form on the phagophore (Figure 9E). The N-terminal IDPR in Atg8 is regulatory and can adopt different conformations depending on the lipidation state of Atg8 [80]. Dimeric Atg8 on the phagophore interacts with Atg40 dimers to promote reticulophagy. This interaction is mediated by a short linear sequence, the AIM motif (LIR motif in higher eukaryotes), in the Atg40 IDPR that binds into two hydrophobic pockets (the W- and L-site) on the surface of the Atg8 UBL domain [58,59,81] (Figure 9E). In reticulophagy receptors, the AIM/LIR is further extended by a C-terminal helix that brings an additional binding affinity of the receptors to Atg8 [79,82]. The protein–protein interaction between the Atg8 and Atg40 assemblies creates forces for ER remodeling, where bending and curving ultimately leads to fragmentation and engulfment of the organelle by the autophagosome (Figure 9E). Fragmentation of the ER membrane is indispensable for the efficient clearance of the endoplasmic reticulum by autophagy [60,79]. Why does the fragmentation rely on the Atg40 disordered region? One possible reason could be that the Atg40 IDPR provides a binding mode with a small footprint on the Atg8 surface, where the disordered conformation is compatible with tight packing of the Atg8- and Atg40-assemblies free of steric clashes, which would occur if a globular domain mediated the Atg8–Atg40 interaction (Figure 9E). A disorder-based binding mode appears to be important because membrane remodeling mediated by a disordered conformation is evolutionary conserved. Human RETREG1, a reticulophagy receptor in mammals, adopts a mechanism similar to that in Atg40 [60,61]. Specifically, the reticulon-homology domain wedges that curve the ER membrane by clustering, as above-mentioned in Section 3, cooperate with the intrinsically disordered C-terminus, which binds to LC3, a human homolog of Atg8 via the LIR motif. Together, they form pulling forces that cause ER vesicle budding, pinching off, and finally fragmentation of the ER.

Another reason for the need of a disordered domain in reticulophagy receptors appears to be the necessity of bridging the space between the ER and the phagophore. TEX264, a single-pass transmembrane receptor for reticulopagy in mammalian cells, is an example of this mechanism [83]. TEX264 consists of an N-terminal transmembrane α-helix, gyrase inhibitor (GyrI)-like domain, and C-terminal disordered domain (residues 193–313), oriented into the cytosol. The very C-terminus binds to LC3 via the LIR motif (residues 273–276 in human protein). Shortening of the TEX264 IDPR by 56 amino acid residues yields a deletion mutant that fails in restoring reticulophagy and co-localization with LC3. In contrast, a mutant shortened by only 32 residues still allows for the receptor function. Furthermore, producing a chimera with the changed amino acid sequence, but retaining length and flexibility by replacing a portion of the TEX264 IDPR with the IDPR from ATG13 does not interfere with reticulophagy. This suggests that the TEX264 IDPR fulfills the function of a flexible bridge in which the length, and not the specific sequence, is the essential feature [83,84].

## 6. Unsolved Mechanisms of Membrane Association of Autophagy Proteins via a Disordered Region

Despite tremendous progress in the elucidation of the IDPs/IDPRs–membrane interface in the autophagy pathway, there is still an array of peripheral membrane proteins that act on autophagy membranes via elusive molecular mechanisms. Current experimental evidence with these proteins indicates that their membrane attachment is regulation prone and/or temporary, suggesting an involvement of an intrinsically disordered region. The first polypeptide in this category is Atg1/ULK1, specifically, the Atg1/ULK1 early autophagy targeting/tethering (EAT) domain, which consists of six α-helices corresponding to two microtubule-interacting and transport (MIT) regions. When probed with liposomes of various sizes, the Atg1 EAT from yeast exhibited liposome tethering and a strong binding ability to highly curved small unilateral vesicles [85]. However, in the presence of protein components of the Atg1 complex, the Atg17–Atg31–Atg29 trimer and Atg13, the Atg1 EAT loses its liposome tethering ability, indicating that the domain may interact with a protein component instead. Indeed, hydrogen-deuterium exchange experiments showed that the C-terminal subdomain of the Atg1 EAT (MIT2) is highly dynamic and can be significantly stabilized by the interaction with Atg13 [86]. How the Atg13-free Atg1 EAT interacts with membranes and how this interaction is regulated by Atg13 remains to be investigated. Given the dependence on membrane curvature, the Atg1 EAT may utilize an amphipathic α-helix. Existence of such a helix and a possible PTM-dependence of its insertion into the lipid bilayer require examination by future studies. Answering these questions is also relevant to Atg38, a fifth subunit of the Vps34 complex in yeast. Atg38, an ortholog of mammalian NRBF2, consists of an N-terminal MIT domain connected via a disordered linker to a C-terminal α-helical domain responsible for homodimerization [87] via the formation of a coiled-coil structure. Subcellular fractionation experiments with Atg38 overexpressed in yeast cells showed that the protein is detected in the membrane associated fraction. Single segment deletions in the C-terminal domain of Atg38 retain the dimeric form, but cause a displacement from the membrane fraction, suggesting that Atg38 may have a membrane-binding domain separate from the dimerizing region at the C-terminus [87]. Whether the Atg38 MIT alone, in analogy to the Atg1 EAT, has a membrane-binding capability is unknown. If the Atg38 MIT interacts with the lipid bilayer, as does Atg1 EAT [85], it remains to be elucidated why and by which mechanism the MIT domains lose this capability after incorporation into a particular protein complex. In other words, the question is whether the MIT domains function as a tool for membrane recruitment and then disengage from this function after assembly into a protein complex.

Autophagosome biogenesis relies on SNARE proteins that mediate the fusion of various cellular membranes [88,89]. Sec9 is an intrinsically disordered peripheral SNARE protein [90] that forms a binary t-SNARE complex with Sso1 in yeast [91], and is important for organizing Atg9 into tubulovesicular structures [92]. Sec9 localizes to the plasma membrane [93] when it is associated with Sso1, which is anchored in the membrane by a transmembrane domain. Specifically, monomeric Sso1 in the “closed” conformation mediated by a four helix bundle switches to an “open” state in the binary complex with Sec9, in which Sso1 contributes one α-helix in the middle of the sequence and Sec9 contributes two N-terminal α-helices. All α-helices in the binary complex are induced from IDPRs. This disorder-to-order transition allows the C-terminal SNARE coiled-coil regions of Sso1 and Sec9 to zipper with Snc1 into a ternary SNARE complex [91]. Biochemical experiments showed that treatment with sodium carbonate (pH 11.5) or 5 M urea, but not a high salt concentration, extracted Sec9 from membranes [93]. These data support the finding [91] that the IDPRs of Sso1 and Sec9 interact via a folded structure. However, monomeric Sec9? SNAP-25, a disordered human homolog of Sec9 (both proteins contain >80% disordered residues), is anchored as a monomer on the presynaptic plasma membrane via a set of palmitoylated cysteines in its disordered region [94,95]. Monomeric Sec9, on the other hand, does not have cysteine residues for palmitoylation. How is a dynamic disordered conformation of monomeric Sec9 [90,91] temporarily constrained before it binds to Sso1? Is there an unknown segment in the disordered Sec9 sequence with a membrane-binding capability that initially recruits the protein to the membrane? The same question stands for human SNAP-29, an intrinsically disordered polypeptide (containing >85% disordered residues) that forms a binary t-SNARE complex with STX17, which binds ATG14 on the autophagosome [96]. Whether a liquid–liquid phase separation combined with fly casting could yield a membrane-independent association of Sec9 and SNAP-29 with their protein partners is a possibility for exploration in future studies.

Another protein that has not been elucidated with respect to membrane binding is Atg17, a subunit of the Atg1 complex in yeast. The N-terminal region of Atg17 (residues 1–115) is well folded, whereas the rest of the sequence comprises a hybrid polypeptide chain mixing structured domains with disordered regions, where one can find coiled-coil domains (residues 146–221 and 353–384) (Figure 10A). The overall disorder content of this protein is ~24%, which classifies it as moderately disordered. Atg17 dimerizes and binds the Atg31–Atg29 heterodimer, which together forms a dimeric heterotrimer with 2:2:2 stoichiometry. The dimeric Atg17–Atg31–Atg29 trimer does not bind liposomes, because Atg31–Atg29 creates a steric bloc [85]. However, Atg17 alone strongly associates with the membrane fraction in subcellular fractionation experiments performed with multiple knock-out yeast cells expressing Atg17-MYC and lacking all proteins from the core autophagy machinery including Atg1, Atg9, Atg13, Atg29, and Atg31 (Figure 10B). These experiments, executed in the same way as subcellular fractionations with Atg16 [54], suggest that a loop in Atg17 functions as a membrane anchor and that the Atg31–Atg29 dimer acts as its negative regulator. Molecular position of a segment in Atg17 that attaches to the lipid bilayer and the mechanism disengaging the protein from the membrane under regulation by the Atg31–Atg29 subunit remain to be investigated. Along these lines, it is noteworthy that Atg17 expressed in multiple knock-out yeast cells associates strongly with membranes (Figure 10B) as does Atg16 probed by the same method. Atg16 utilizes an amphipathic α-helix for a strong attachment to the membrane [54]. It is possible that exclusively helical Atg17 (Figure 10C) applies a similar mechanism.

Atg17 in yeast is an essential protein in the initiation of bulk autophagy under nutrient-deficient conditions, but is dispensable in the Cvt pathway under growing conditions [97], where the peripheral membrane-binding protein Atg23 is essential [98]. In contrast to Atg17, the Atg23 amino acid sequence grants the protein a significant structural flexibility with the overall disorder content exceeding 70% (Figure 11A). A recent study [99] revealed that Atg23 has an elongated α-helical fold. This finding is in agreement with the AlphaFold model (Figure 11B), which also shows that the Atg23 molecule was enriched on the surface with positively charged amino acid residues. The protein likely utilizes these positive charges to approach phospholipids because the membrane-binding capability of Atg23 is sensitive to ionic strength in buffer [99]. Atg23 dimerizes via an amphipathic α-helix (residues 171–189, *S. cerevisiae* numbering) and tethers liposomes, suggesting that the protein should apply, in addition to electrostatic forces, a hydrophobic insertion that pulls vesicles. An Atg23 mutant that is unable to dimerize exhibits a decreased α-helical content, is partially deficient in membrane binding, and fails in vesicle tethering [99]. This suggests, in agreement with bioinformatics analysis (Figure 11A), that a disordered loop in an extended conformation (residues 150–250) undergoes a disorder-to-order transition upon dimerization, which forms the AH and concomitantly positions an unknown set of residues for membrane binding in cooperation with positive surface charges, making dimerization and vesicle tethering by Atg23 spatiotemporally coupled events (Figure 11C). How this coupling mechanism is orchestrated and where a hydrophobic membrane anchor is positioned in the Atg23 molecule remains to be investigated. Independence of membrane binding on membrane curvature, which was observed for Atg23, would indicate that this anchor is not a strong AH. More experiments are needed to examine whether protonation of glutamate/aspartate side chains plays a role in its hydrophobicity and membrane affinity. It is possible that solving these questions for Atg23 will provide a clue for the mechanism of Atg17–lipid attachment, as these two proteins resemble each other in the overall architecture of elongated α-helical antiparallel dimers, and both interact with Atg9, a transmembrane protein embedded in small lipid vesicles [100,101] (Figure 1).

## 7. Conclusions

An ultimate goal of the autophagy pathway is building, de novo, a membrane organelle, the autophagosome, where the cargo is degraded. During this process, many proteins act on the growing phagophore in a temporary and highly regulated fashion, which is a hallmark of protein intrinsic disorder. Therefore, it is not surprising that building the autophagosome requires a significant intertwining between various membranes and intrinsically disordered proteins or protein regions of the autophagy machinery. In this review, and to the best of our knowledge, we provided an overview of all interactions, effects, or mutually shaping processes that are currently recognized to take place in the autophagy pathway at the IDPs/IDPRs–membrane interface. We can only wish that there are more reported studies to include, as we are aware that this summary only touches on the magnitude and complexity of these events. Many autophagy proteins containing IDPRs still lack elucidated structures. Our hope is that as we gain more insights into the structure–function relationships of these proteins, we disclose a bigger portion of the “iceberg” that is underneath the tip discussed in this review.

## Figures and Tables

**Figure 1 membranes-12-00457-f001:**
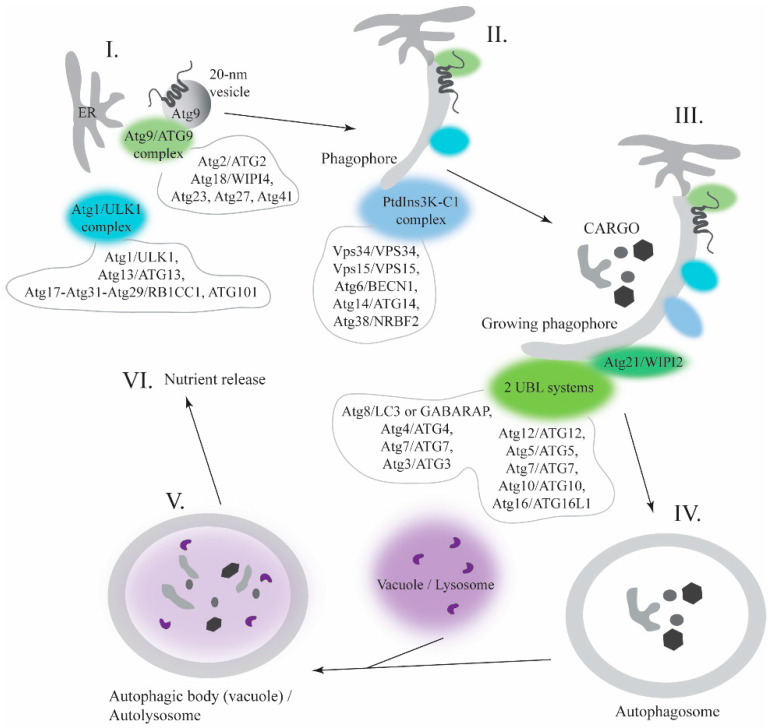
Schematic representation of the autophagy pathway. (**I**) Induction: Atg9/ATG9 vesicles, components of the Atg9 (yeast)/ATG9 (higher eukaryotes) complex, the endoplasmic reticulum (ER), and the Atg1/ULK1 complex gather together. (**II**) Nucleation: The PtdIns3K-CI complex enriches the phagophore phospholipid membrane with PtdIns3P, which leads to the recruitment of additional components of the autophagy machinery. (**III**) Expansion: Recruitment of the two ubiquitin-like (UBL) protein systems promotes conjugation of Atg8/LC3 or GABARAP to phosphatidylethanolamine (PE), and ensures the phagophore expansion. (**IV**) Maturation: Closure of the phagophore engulfs the cargo and forms the double-membrane autophagosome. Autophagy components leave the outer membrane and the mature autophagosome fuses with the vacuole/lysosome. (**V**) Breakdown: A large size of the vacuole leads to the release of the inner autophagic vesicle, the autophagic body, into the vacuolar lumen. In higher eukaryotes, a small lysosome fused with the autophagosome forms the autolysosome. Resident hydrolases and lipases degrade the cargo. (**VI**) Release: Resulting macromolecules are released into the cytosol for reuse of nutrients.

**Figure 2 membranes-12-00457-f002:**
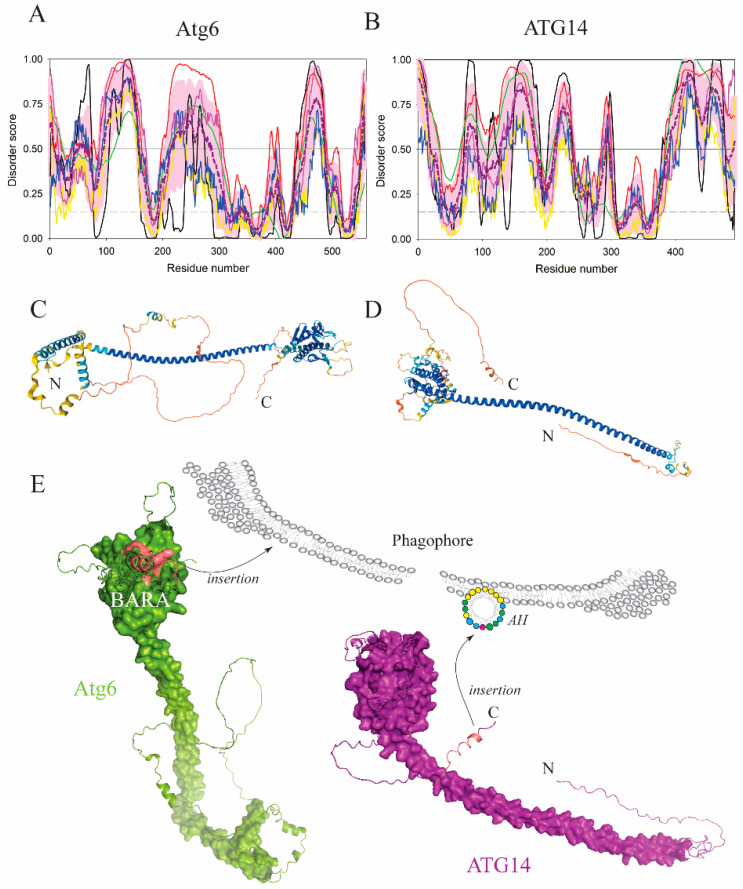
The Atg6–Atg14/BECN1–ATG14 complex in yeast and mammals binds to membranes. (**A,B**) Bioinformatics analysis of the amino acid sequences of the Atg6 in yeast (*S. cerevisiae)* and human ATG14 shows that N-terminal coiled-coil domains in each protein are intrinsically disordered. This analysis was conducted using RIDAO web crawler, which was designed for the rapid prediction and comparison of protein disorder profiles. It aggregates the results from a number of well-known disorder predictors: PONDR^®^ VLXT [21] (black line), PONDR^®^ VL3 [22] (green line), PONDR^®^ VLS2 [23] (red line), PONDR^®^ FIT [24] (pink line), IUPred2 (Short), and IUPred2 (Long) [25,26] (yellow and blue line, respectively). For each query protein, we also generated mean disorder profile (MDP, dashed dark pink line) by averaging the outputs of individual predictors and calculated the corresponding errors (shown as light pink shadow). The outputs of the evaluation of the per-residue disorder propensity by these tools are represented as real numbers between 1 (ideal prediction of disorder) and 0 (ideal prediction of order). A threshold of ≥0.5 was used to identify disordered residues and regions in query proteins, whereas residues with disorder scores ranging from 0.15 to 0.5 were considered as flexible. The C-terminal BARA domain in Atg6 contains a disordered loop and the C-terminal BATS region in ATG14 is an IDPR. As per the results of the PONDR^®^ VLS2 analysis, the overall disorder contents of Atg6 and ATG14 were 55.8% and 61.2%, respectively. According to the accepted practice, proteins can be classified based on their percent of predicted disordered residues (PPDR) as highly ordered (PPDR < 10%), moderately disordered (10% ≤ PPDR < 30%), and highly disordered (PPDR ≥ 30%) [27]. Based on these criteria, both Atg6 and ATG14 are highly disordered. (**C,D**) The AlphaFold model of Atg6 and ATG14 structures with color-coded modeling confidence. Very high confidence, blue; high confidence, cyan; medium confidence, yellow; low confidence, orange. AlphaFold is a deep learning-based approach, which is the most advanced and accurate computational tool for predicting protein structure [28]. This novel method is revolutionizing protein structural predictions. Furthermore, it was indicated that many AlphaFold-generated model structures contain regions of low and very low confidence, which often overlap with IDPRs [29]. However, it was also pointed out that in many structures modeled by the AlphaFold, IDPRs are represented as unrealistic structures, being shown as large loops clashing with the overall protein geometry [30]. Furthermore, the AlphaFold fails to provide an accurate description of the conformational ensembles of IDPRs [29], as a single static structure cannot be used to reflect the relative movements of residues in such structural ensembles. Finally, fully disordered proteins or fully disordered regions cannot be accurately identified by the AlphaFold, which almost always predicts some residual structure [31]. Therefore, in this and subsequent figures, the AlphaFold is exclusively used to generate protein 3D structures for illustrative purposes. (**E**) Protein surface rendering of the Atg6 and ATG14 models from C and D. The membrane-binding segment in the Atg6 BARA and the amphipathic helix (AH) at the C-terminus of ATG14, which insert into the phagophore membrane, are highlighted in dark pink. The schematic representation of the AH shows the hydrophobic face in yellow and the hydrophilic face in green, blue, and red depicting polar, positively charged, and negatively charged amino acid residues, respectively.

**Figure 3 membranes-12-00457-f003:**
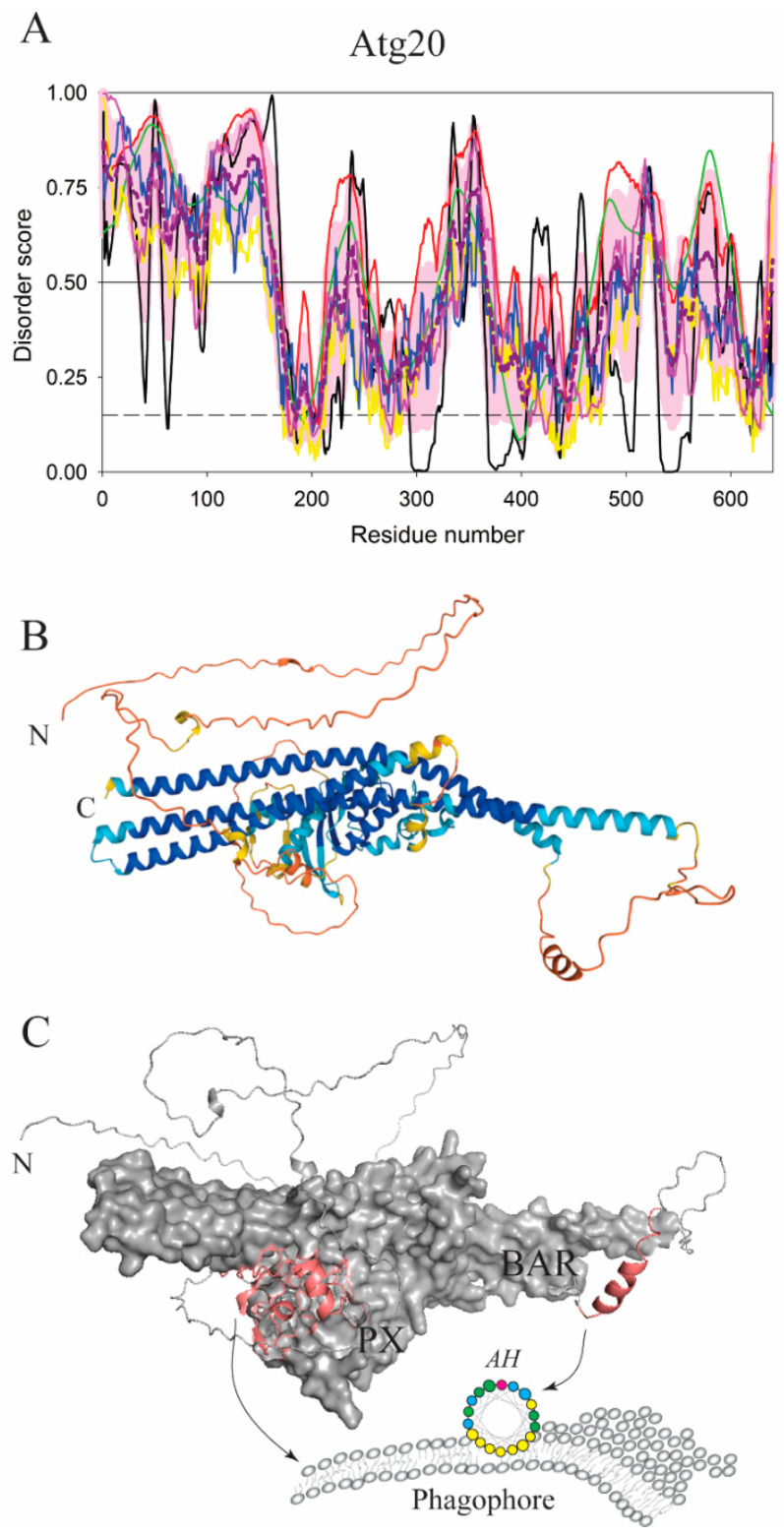
Atg20 is a structurally unique Phox homology-Bin/Amphiphysin/Rvs (PX-BAR) sorting nexin in autophagy. (**A**) Bioinformatic analysis of the Atg20 from yeast *S. cerevisiae* (for keys, see legend to Figure 2) showed a disordered score for the intrinsically disordered N-terminus (residues 1–160), a flexible loop called the PX-GAP in the PX domain (residues 161–297), the disordered linker between the PX and BAR domains (residues 298–358), and the unstructured BAR-GAP in the BAR domain (residues 359–636). Based on its overall disorder content of 67.3%, Atg20 belongs to the category of highly disordered proteins. (**B**) The AlphaFold model of Atg20 with color-coded modeling confidence. Very high confidence, blue; high confidence, cyan; medium confidence, yellow; low confidence, orange. (**C**) Protein surface rendering of the Atg20 model from B. The structured regions of the PX and BAR domain are denoted in black, and the membrane-binding segments in the PX and BAR domains are highlighted in dark pink.

**Figure 4 membranes-12-00457-f004:**
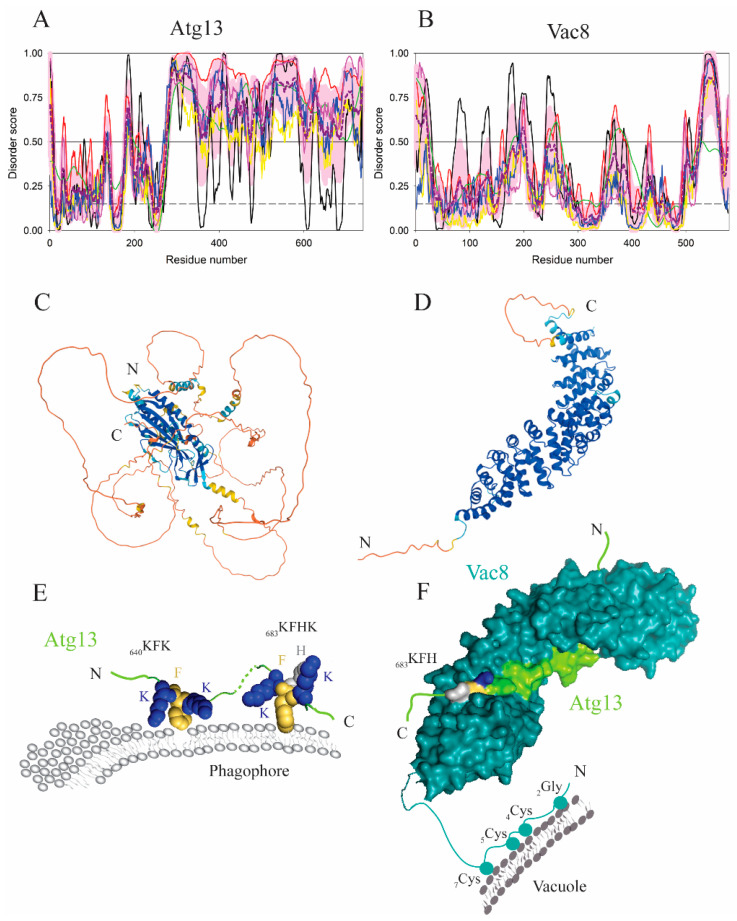
Subunits of the Vac8–Atg13 complex bind to membranes. (**A,B**) Bioinformatics analysis of the amino acid sequences of Atg13 and Vac8 from yeast, *S. cerevisiae*, showed a high disorder score for a long intrinsically disordered region in Atg13 as well as for the flexible N-terminal tail and the C-terminal loop in Vac8. As per the results of the PONDR^®^ VLS2 analysis, the overall disorder contents of Atg13 and Vac8 were 71.5% and 27.7%, respectively, classifying them as highly (Atg13) and moderately disordered (Vac8) proteins. (**C**,**D**) The AlphaFold model of Atg13 and Vac8 with color-coded modeling confidence. Very high confidence, blue; high confidence, cyan; medium confidence, yellow; low confidence, orange. (**E**,**F**) Membrane-binding mechanism for Atg13 and Vac8. The membrane-binding motifs, _640_KFK and _683_KFHK in Atg13 employ electrostatic and hydrophobic forces to interact with the phagophore membrane. Posttranslational modifications, myristoylation and palmitoylation, of glycine and cysteine residues at the N-terminus of Vac8 anchor the protein to the vacuolar membrane. Vac8 is visualized with Atg13 bound to its cationic groove (PDB ID: 6KBM).

**Figure 5 membranes-12-00457-f005:**
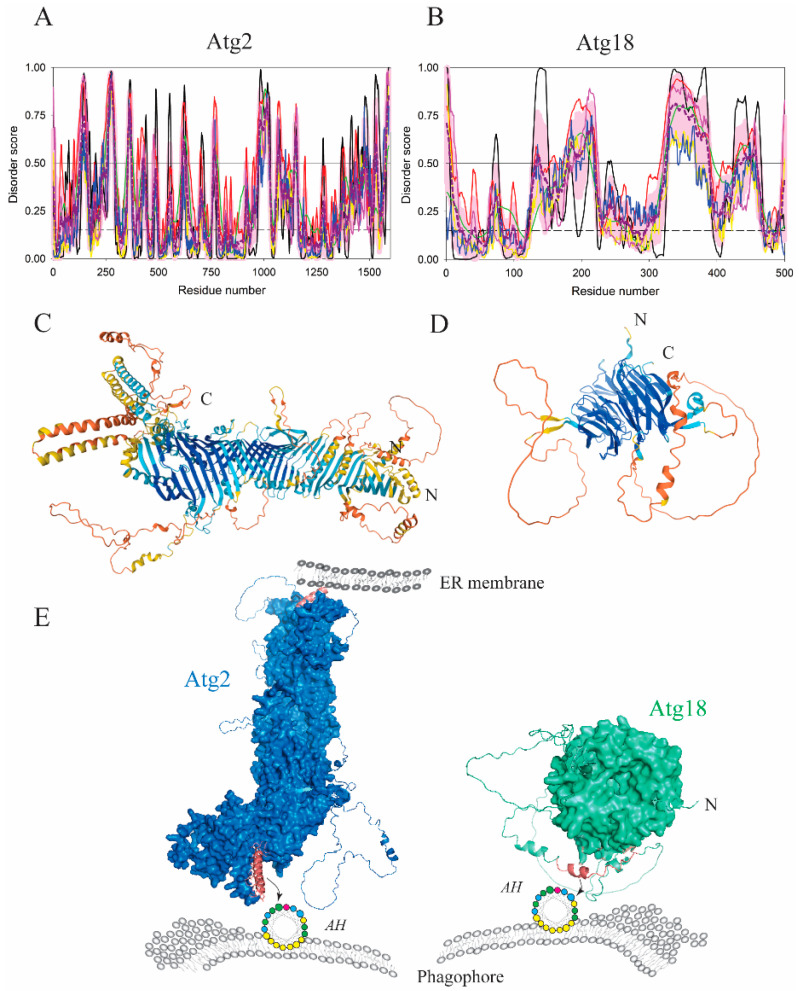
The Atg2–Atg18 complex is anchored to the phagophore via an amphipathic helix in each protein. (**A,B**) Bioinformatics analysis of the Atg2 and Atg18 proteins from yeast, *S. cerevisiae*, showed, based on the disorder score, that Atg2 is a well-folded protein with many flexible loops, whereas the Atg18 β-propeller is significantly interrupted by gaps filled by disordered regions. PONDR^®^ VLS2-based analysis revealed that the overall disorder contents of Atg2 and Atg18 were 29.9% and 38.8%, respectively, indicating that these two proteins are moderately disordered. (**C,D**) The AlphaFold model of Atg2 and Atg18 with color-coded modeling confidence. Very high confidence, blue; high confidence, cyan; medium confidence, yellow; low confidence, orange. (**E**) Protein surface rendering of the Atg2 and Atg18 models from C and D. The membrane-binding segments (dark pink) transition into the amphipathic α-helices in Atg2 and Atg18 that insert into the phagophore membrane. The N-terminal tip of Atg2 attaches to the ER membrane, making Atg2 a lipid transporter from the ER to the phagophore.

**Figure 6 membranes-12-00457-f006:**
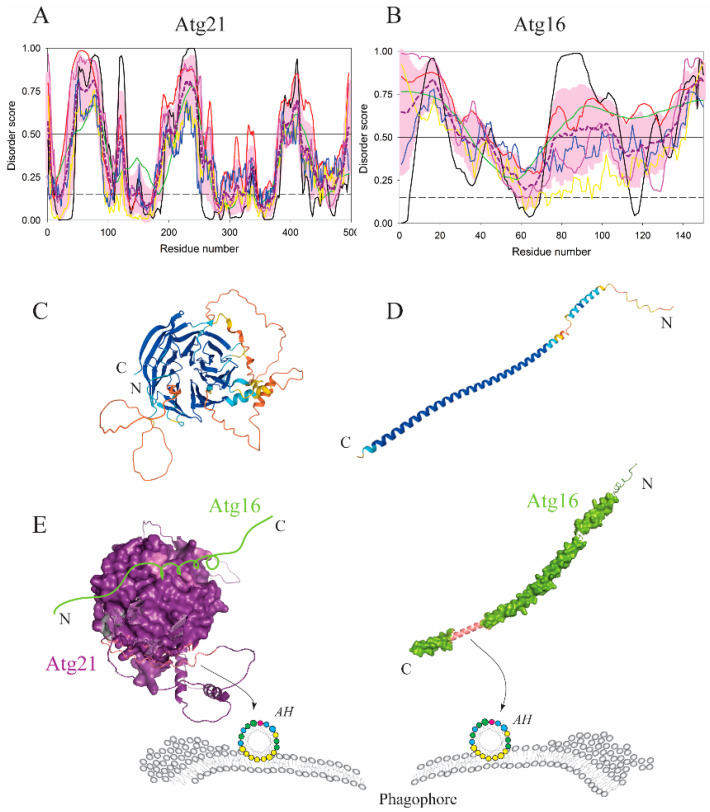
Atg21 and Atg16 in yeast utilize amphipathic helices for their association with the phagophore. (**A**,**B**) Disorder–order prediction from the amino acid sequence analysis of Atg21 and Atg16 from yeast, *S. cerevisiae*, showed that Atg21 is a hybrid protein with numerous disordered regions protruding from a folded architecture, whereas Atg16 is an intrinsically disordered protein in its monomeric form. According to the PONDR^®^ VLS2-based analysis, Atg21 and Atg16 are classified as highly disordered proteins, since their overall disorder contents amounted to 44.35% and 80.7%, respectively. (**C**,**D**) The AlphaFold model of Atg21 and Atg16 with color-coded modeling confidence. Very high confidence, blue; high confidence, cyan; medium confidence, yellow; low confidence, orange. Atg21 folds into an IDPR-enriched β-propeller, whereas Atg16 is a coiled-coil containing protein that undergoes several disorder-to-order transitions yielding mostly a helical structure upon dimerization and interaction with physiological binding partners. One monomer after these transitions is depicted. (**E**) Protein surface rendering of the Atg21 and Atg16 models from C and D. The membrane-binding segments that give rise to the amphipathic α-helices are highlighted in dark pink. Atg21 recruits the disordered Atg16 monomer to the phagophore, where Atg16 binds via its C-terminal AH.

**Figure 7 membranes-12-00457-f007:**
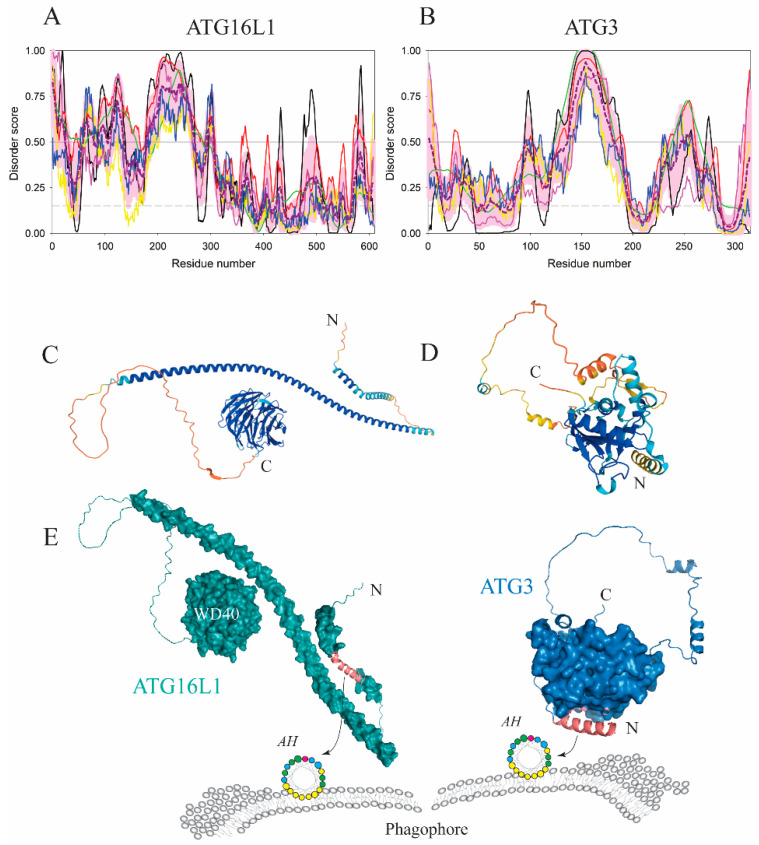
Human ATG16L1 and ATG3 employ amphipathic α-helices for association with the phagophore. (**A**,**B**) Bioinformatics analysis of the amino acid sequences of ATG16L1 and ATG3 showed that ATG16L1 consists of the flexible N-terminus and well-folded C-terminus, whereas the ATG3 architecture is disrupted by a large disordered loop. In fact, PONDR^®^ VLS2 indicates that with their overall disorder contents of 51.1% and 34.4%, ATG16L1 and ATG3 belong to the categories of highly and moderately disordered proteins, respectively. (**C**,**D**) The AlphaFold model of ATG16L1 and ATG3 with color-coded modeling confidence. Very high confidence, blue; high confidence, cyan; medium confidence, yellow; low confidence, orange. (**E**) Protein surface rendering of the ATG16L1 and ATG3 models from C and D. The membrane-binding segments that form the amphipathic α-helices are highlighted in dark pink. The C-terminus of ATG16L1 folds into a WD40 domain and the protein dimerizes via a coiled-coil region located upstream of the disordered linker with WD40.

**Figure 8 membranes-12-00457-f008:**
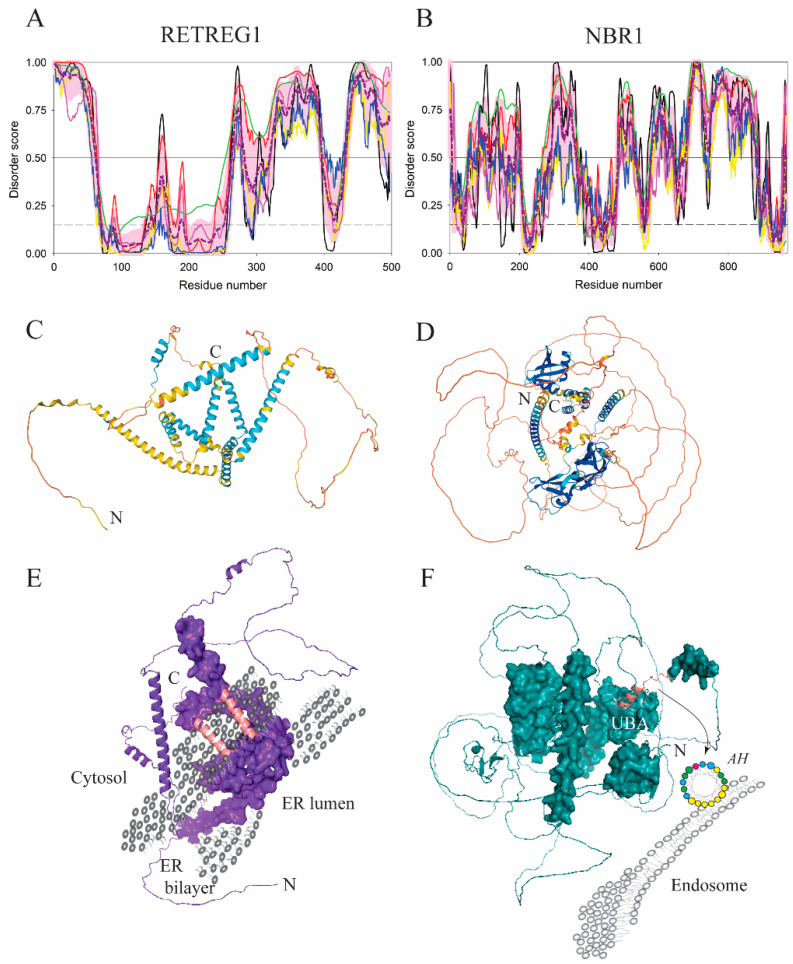
Intrinsic disorder in autophagy receptors. (**A**,**B**) Bioinformatics analysis of the amino acid sequences of RETREG1 and NBR1 showed that RETREG1 consists of the flexible N-terminus followed by a well-folded domain and long highly disordered C-terminal domain, whereas the NBR1 contains multiple IDPRs of different lengths. As per the results of the PONDR^®^ VLS2 analysis, the overall disorder contents of RETREG1 and NBR1 were 56.3% and 62.1%, respectively, classifying both of them as highly disordered proteins. (**C**,**D**) The AlphaFold models of the RETREG1 and NBR1 structures with color-coded modeling confidence. Very high confidence, blue; high confidence, cyan; medium confidence, yellow; low confidence, orange. (**E**,**F**) Protein surface rendering of the RETREG1 and NBR1 models in C and D. The RETREG1 reticulon-homology domain forms a wedge-shaped structure that curves membranes. This structure is embedded in the ER membrane, where two amphipathic helices in RETREG1 (dark pink) are inserted into the bilayer on the cytosolic side. The intrinsically disordered N- and C-termini of RETREG1 protrude into the cytosol. The NBR1 complex architecture is significantly enriched in IDPRs, and carries an ubiquitin-associated (UBA) domain at the C-terminus. A short α-helix preceding the UBA is amphipathic (dark pink) and inserts into the lipid bilayer of the late endosomes.

**Figure 9 membranes-12-00457-f009:**
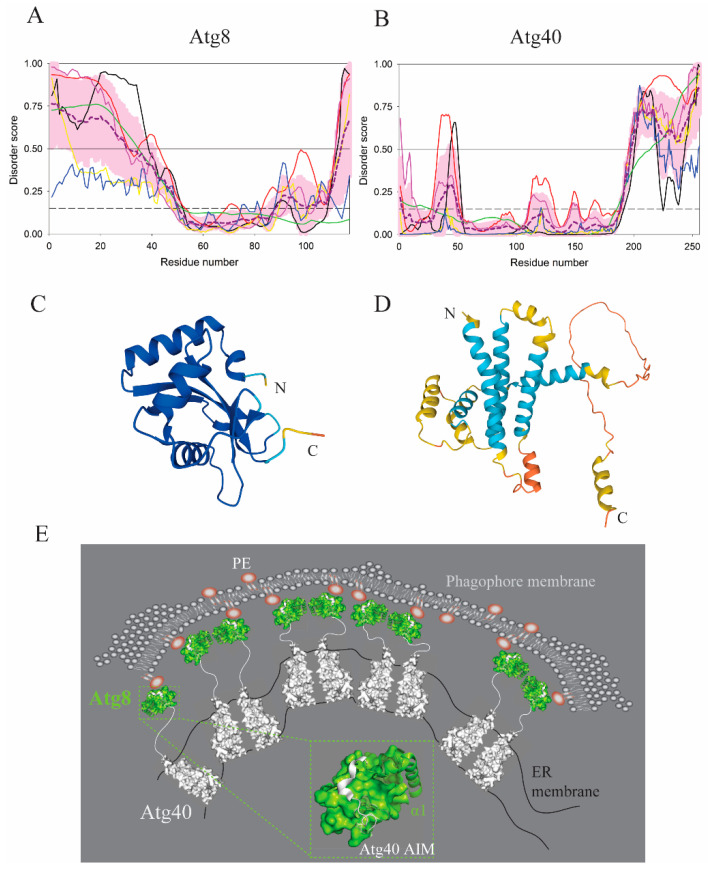
ER-membrane remodeling induced by the Atg40–Atg8 interaction. (**A**,**B**) Bioinformatics analysis of the amino acid sequences of Atg40 and Atg8 showed that the long IDPRs are located at different sides of these proteins, with the C- and N-terminal regions of Atg40 and Atg8 being disordered, respectively. Based on their overall disorder contents of 28.9% and 37.6%, Atg40 and Atg8 are classified as moderately disordered. (**C**,**D**) The AlphaFold models of the Atg40 and Atg8 structures with color-coded modeling confidence. Very high confidence, blue; high confidence, cyan; medium confidence, yellow; low confidence, orange. (**E**) Schematic visualization of the membrane-remodeling mechanism. Atg8 (green) conjugated to PE on the phagophore membrane dimerizes and assembles. Dimeric Atg40 (white) in the ER membrane binds Atg8 via the intrinsically disordered C-terminal region. Puling forces induced by the Atg40–Atg8 interaction remodel the ER in order to fragment and pack this organelle into the autophagosome. A close up view of the Atg40–Atg8 interacting interface is depicted in the green dash-lined rectangle. The AIM motif along with the α-helical element in Atg40 (white) bound to the surface of the Atg8 UBL (green) provides a strong binding affinity that facilitates bending and curving of the ER membrane. The regulatory N-terminus in Atg8, depicted as the α1 helix in ribbon representation, adopts an unknown conformation that supports dimerization, and may not be α-helical in lipidated Atg8 assemblies.

**Figure 10 membranes-12-00457-f010:**
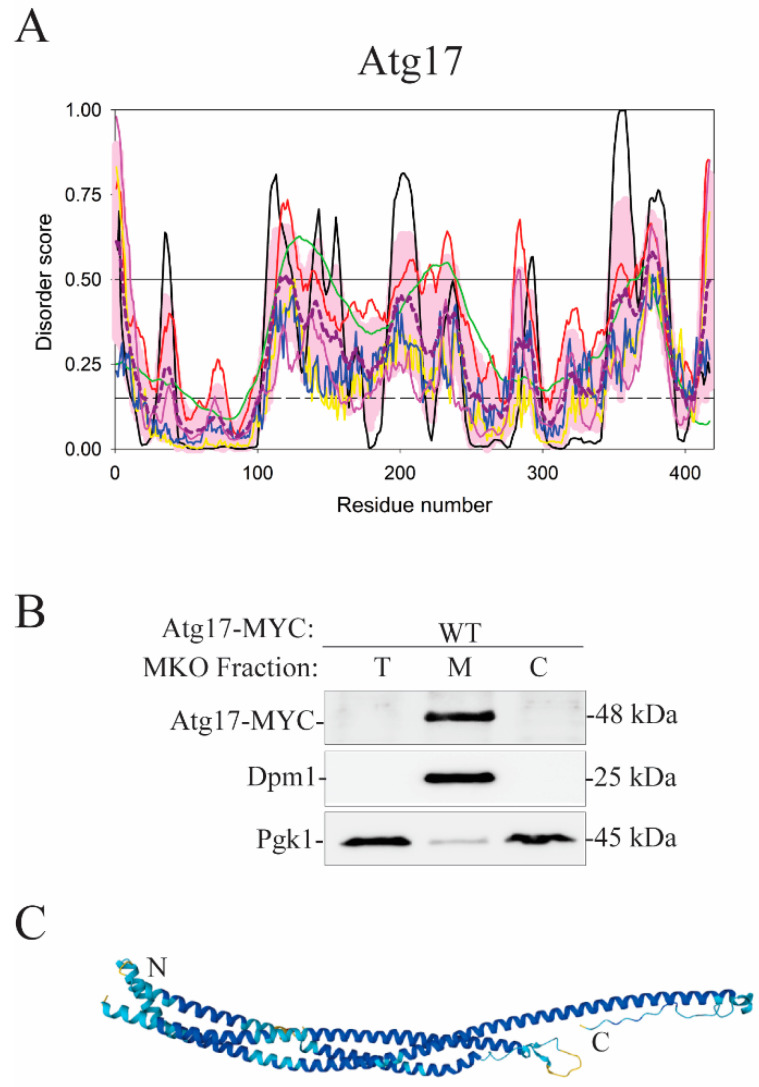
Atg17 is a putative membrane-binding protein. (**A**) Bioinformatics analysis of the Atg17 amino acid sequence from yeast *S. cerevisiae* showed protein segments with a higher flexibility indicated by the disorder scores above the black line, a threshold for protein intrinsic disorder. These segments are candidates for a membrane-binding domain in Atg17. With the overall disorder content of 23.7%, Atg17 belongs to the class of moderately disordered proteins. (**B**) Atg17 strongly associates with membranes based on subcellular fractionation of multiple knock-out (MKO) yeast cells expressing Atg17-MYC. All known Atg17-binding proteins are deleted in this cell line. T, total fraction; M, membrane-associated fraction; C, cytosolic fraction. Pgk1 and Dpm1 were used as controls representing a cytosolic and membrane-associated protein, respectively. (**C**) The AlphaFold model of monomeric Atg17 with color-coded modeling confidence. Very high confidence, blue; high confidence, cyan; medium confidence, yellow; low confidence, orange.

**Figure 11 membranes-12-00457-f011:**
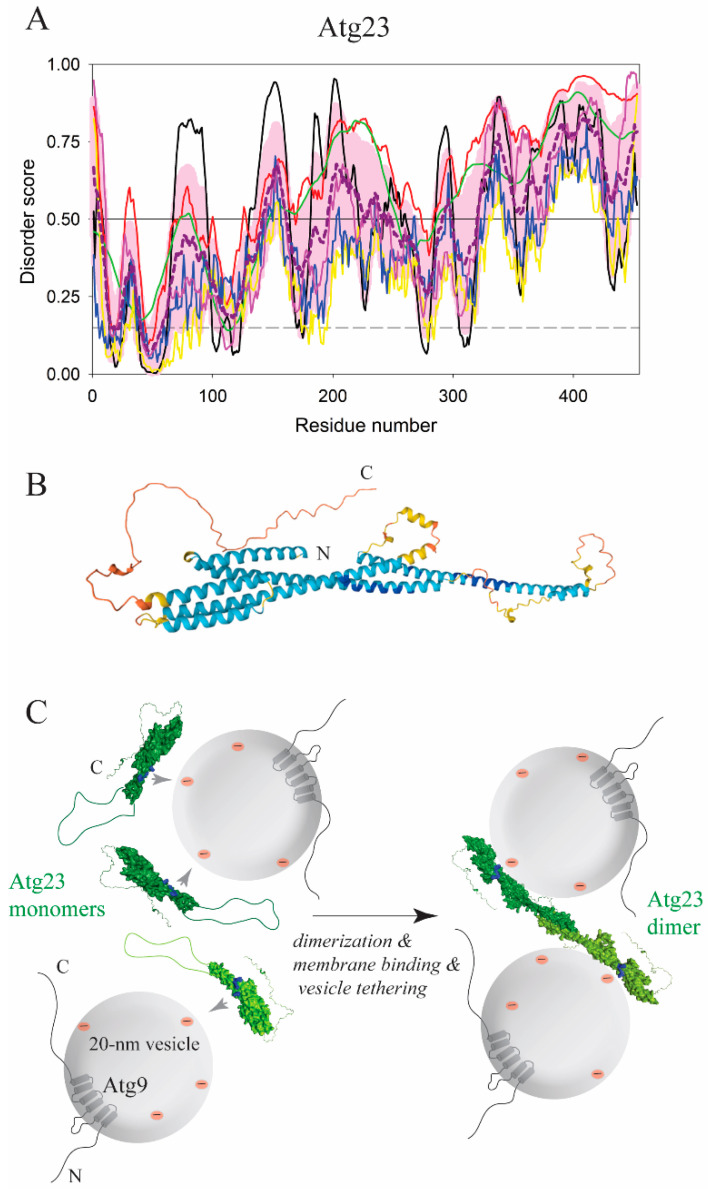
Disorder-to-order transition in Atg23 from yeast mediates dimerization, membrane binding, and vesicle tethering. (**A**) Bioinformatics analysis of the amino acid sequence from Atg23 showed that the protein possesses a disordered loop in the region between amino acid residues 150–250 and the intrinsically disordered C-terminus. With 73.7% residues predicted to be disordered, Atg23 clearly belongs to the category of highly disordered proteins. (**B**) The AlphaFold model of Atg23 with color-coded modeling confidence. Very high confidence, blue; high confidence, cyan; medium confidence, yellow; low confidence, orange. (**C**) Schematic visualization of putative structural transitions in Atg23. Positive surface charges in monomeric Atg23 are attracted to negatively-charged Atg9-containing phospholipid vesicles (*left*). Proximity of protein molecules leads to dimerization of Atg23 concomitantly with additional membrane binding and vesicle tethering (*right*). These three processes are spatiotemporally coupled.

**Table 1 membranes-12-00457-t001:** Autophagy proteins engaging their intrinsically disordered protein regions (IDPRs) in actions on membranes via various molecular mechanisms. The percent of predicted disordered residues (PPDR) defines the proteins as highly ordered (PPDR < 10%), moderately disordered (10% ≤ PPDR < 30%), or highly disordered (PPDR ≥ 30%). Note that none of the analyzed proteins could be classified as highly ordered. PPDR values were retrieved from the results of PONDR^®^ VSL2 analysis. Names of moderately and highly disordered proteins are shown by pink and red font, respectively. Long IDPRs (i.e., IDPRs containing at least 30 residues) are shown by the **bold red** font. The question mark represents an unknown or putative mechanism/posttranslational modification (PTM) that needs to be elucidated.

Protein (length)	PPDR (%)	Positions of IDPRs	Act with Membranes	Molecular Mechanism MEDIATED by IDPR	PTM related to the Molecular Mechanism
Atg1 (897)	42.8	1–17, 26–29, 45–53, 57–65, 119–121, 133–142, **328–563**, 583–585, **628–683**, 770–787, 854–864, 890–897	Tethering	?	?
Atg2 (1592)	29.9	35–41, 96–102, **112–161**, 195–196, **230–296**, 352–375, 399–405, 410–436, 474–491, 607–643, 697–704, 757–779, 914–919, 949–952, **964–1026**, 1116–1121, 1145–1159, 1193–1196, 1429–1442, 1478–1485, 1527–1536, 1573–1592	Binding	AH	Dephosphorylation?
ATG3 (314)	34.4	1–5, 25–30, 97–101, **129–180**, 183–191, 235–250, 309–313	Binding	AH	?
Atg6 (557)	55.8	1–6, 9–20, 38–40, 42–61, 73–80, **88–162**, **199–307**, 389–395, 399–406, 435–440, **446–490**, 507–508, 539–-540, 550–557	Binding	Disordered loop	?
Atg8 (117)	37.6	**1–30**, 35–42, 112–117	Lipidation	PE conjugation	C-terminal cleavage
Atg13 (738)	71.5	1–6, 30–37, 96–97, 129–143, 177–193, 204–212, 229–232, **272–738**	Binding	Lys/Phe-enriched motif	Dephosphoryation?
ATG14 (492)	61.2	**1–33**,**68–181**, **212–242**, 285–298, **384–492**	Binding	AH	Dephosphorylation?
Atg16 (150)	80.7	**1–43**, **73–150**	Binding	AH	?
ATG16L1 (607)	51.1	**1–30**, 43–48, **53–165**, **176–285**, 289–307, 359–369, 407–408, 570–590	Binding	AH	?
Atg17 (417)	23.7	1–6, 114–130, 137–141, 201–212, 218–223, 227–238, 280–288, 352–359, 365–382, 412–417	Binding	?	?
Atg18 (500)	38.8	1–9, 130–138, 153–163, **172–221**, **316–391**, 408–409, **427–459**, 497–500	Binding	AH	Dephosphorylation?
Atg20 (640)	67.3	**1–167**, **213–248**, 256–262, **300–376**, 391–397, 417, 430–434, **474–540**, **550–607**, 635–640	Binding/tubulation	Disordered loop/AH	Dephosphorylation?
Atg21 (496)	44.4	1–4, **30–87**, 117–123, **186–254**, 264–271, 330–334, 336, 338–339, **380–438**, 491–496	Binding	AH	Dephosphorylation?
Atg23 (453)	73.7	1–7, 28–35, 73–85, 100, 126–127, **139–169**, **171–273**, **284–453**	Tethering	Charged residues/?	?
Atg40 (256)	28.9	33–45, **196–256**	Remodeling	AIM for Atg8	Phosphorylation?
RETREG1 (497)	56.3	**1–63**, 157–165, **260–400**, 431–497	Binding/remodeling	AH/LIR for LC3	Phosphorylation?
Vac8 (578)	27.7	1–24, 158–161, 195–203, 242–268, 361–389, 430–435, 496–503, **518–566**, 576–578	Binding	PTM anchor	G2 myristoylation, C4, C5, C7 palmitoylation
NBR1 (966)	62.1	1–5, **52–204**, **267–366**, **481–531**, **581–643**, **662–886**, 961–966	Binding	AH	Dephosphorylation?
RAB33B (229)	35.8	1–26, **174–229**	Binding	PTM anchor	C227, C229 prenylation
YKT6 (198)	27.8	1–4, 49–59, 130–143, 158–178, 187, 195–199	Binding	PTM anchor	C194 palmitoylation, C195 prenylation
TEX264 (313)	37.4	1–4, 40, 100–110, **213–313**	Space bridging	LIR for LC3	Phosphorylation?

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
