# Peer review of "Theater in the Self-Cleaning Cell: Intrinsically Disordered Proteins or Protein Regions Acting with Membranes in Autophagy"

_membranes, 2022, doi:10.3390/membranes12050457_

Round 1

Reviewer 1 Report

The authors provide a thorough review on the disorder-based interactions with membranes in the autophagy machinery.  The manuscript is well written.  I recommend the manuscript to publish in Membranes with my minor comments below.

In section 2 and 5, random coil was mentioned multiple times.  Even though the averaging behaviors of a variety of IDPs can be close to a random coil, for specific IDPs, there often exists different structural features (e.g. local secondary preference or even long-range interactions) within an ensemble of disordered conformations.  It might be better to change the wording to disordered conformations, since random coil is defined as a Gaussian chain in polymer physics.

The legends in the subfigure A and B of many figures are too small to read.  It's also unclear what regions are disordered with so many disorder predictors presented.  It might help by highlighting the disordered region using bold lines with a criterion from combined efforts of multiple predictors like PONDR does.  The authors can also adapted the AlphaFold structured region or confidence into disordered score - residue number plot, since it is increasingly acknowledged AlphaFold2 can be used as a disordered predictor.

A table of all the proteins discussed, a summary of their fraction of disordered region from disordered predictors and a note of their membrane binding mechanism might be helpful.  A table of different types of PTM and their roles might also be helpful.

I got a little bit confused when reading section 2 and 3, since in both sections the role of net charge and hydrophobic amino acids are discussed except that in section 3, there is an order-to-disorder transition so that key amino acids are solvent exposed.  Is that there is no order-to-disorder transition in the proteins discussed in section 2?

In section 6, a variety of phenomena that have yet been explained were presented.  Is there any hypothesis for some of these cases from the authors' perspective?  Is their any known feature in the IDP field that might be used for understanding these phenomena (e.g. IDP allostery, fuzzy complex and etc.)?

Author Response

We thank the reviewer for the recommendation and specific comments.

The authors provide a thorough review on the disorder-based interactions with membranes in the autophagy machinery.  The manuscript is well written.  I recommend the manuscript to publish in Membranes with my minor comments below.

In section 2 and 5, random coil was mentioned multiple times.  Even though the averaging behaviors of a variety of IDPs can be close to a random coil, for specific IDPs, there often exists different structural features (e.g. local secondary preference or even long-range interactions) within an ensemble of disordered conformations.  It might be better to change the wording to disordered conformations, since random coil is defined as a Gaussian chain in polymer physics.

RESPONSE. Thank you for pointing this out. We have replaced the words “random-coil” with “disordered” or “extended disordered”, depending on the context.

The legends in the subfigure A and B of many figures are too small to read.  It's also unclear what regions are disordered with so many disorder predictors presented.  It might help by highlighting the disordered region using bold lines with a criterion from combined efforts of multiple predictors like PONDR does.  The authors can also adapted the AlphaFold structured region or confidence into disordered score - residue number plot, since it is increasingly acknowledged AlphaFold2 can be used as a disordered predictor.

RESPONSE. Thank you for this useful point. We removed the legends from the A(B) panels of all figures. To show disordered regions in proteins we introduce a new table presenting the disordered status and positions of disordered regions in the proteins analyzed in this review.

A table of all the proteins discussed, a summary of their fraction of disordered region from disordered predictors and a note of their membrane binding mechanism might be helpful.  A table of different types of PTM and their roles might also be helpful.

RESPONSE. We appreciate this suggestion. The new Table 1 summarizes the autophagy proteins containing IDPRs that are involved in various molecular mechanisms with membranes. The fraction of disordered regions in each protein is expressed as the percentage of predicted disordered residues (PPDR). PTMs directly related to these mechanisms are also listed in the table, but many PTM regulations have not been elucidated yet. Membrane-unrelated PTMs of autophagy proteins were reported elsewhere (e.g. Xie et al. (2015) Autophagy 11: 20-45; Popelka & Klionsky (2015) FEBS J 282: 3474-3488; Licheva et al. (2022) Autophagy 18: 104-123), as this topic is quite complex, and, due to its extent, is out of the scope of this review.  

I got a little bit confused when reading section 2 and 3, since in both sections the role of net charge and hydrophobic amino acids are discussed except that in section 3, there is an order-to-disorder transition so that key amino acids are solvent exposed.  Is that there is no order-to-disorder transition in the proteins discussed in section 2?

RESPONSE. We thank the reviewer for this useful point and apologize for the confusion. The section 2 describes the membrane binding mechanisms that are associated with unfolded disordered conformation lacking a transient secondary structure. We assume that these extended conformations are always unfolded, but we cannot exclude a possibility that some of them originated in an order-to-disorder transition. We modified the text on page 4 to read “When protein domains, often loops, in extended (unfolded, lacking a transient secondary structure) disordered conformations …”. In contrast, section 3 is focused on transient disorder-to-order mechanisms, mainly formation of α-helical structures during membrane binding.

In section 6, a variety of phenomena that have yet been explained were presented.  Is there any hypothesis for some of these cases from the authors' perspective?  Is their any known feature in the IDP field that might be used for understanding these phenomena (e.g. IDP allostery, fuzzy complex and etc.)?

RESPONSE. Thank you for pointing this out. In the section 6, it is difficult to ascribe an unknown mechanism to a known IDP feature. We tried to refrain from speculation in this section. To address this comment, we added a few hypotheses in the section 6 that bring our perspective. New sentences on page 20 now read: “Given the dependence on membrane curvature, the Atg1 EAT may utilize an amphipathic α-helix. Existence of such a helix and a possible PTM-dependence of its insertion into the lipid bilayer need examination by future studies. Answering these questions is relevant also to Atg38, a fifth subunit of the Vps34 complex in yeast.”

A new text on page 21 now reads: “Whether a liquid-liquid phase separation combined with fly casting could yield a membrane-independent association of Sec9 and SNAP-29 with their protein partners is a possibility for exploration by future studies”.

A new text on page 22 now reads: “Independence of membrane binding on membrane curvature, which was observed for Atg23, would indicate that this anchor is not a strong AH. More experiments are needed to examine whether protonation of glutamate/aspartate side chains plays a role in its hydrophobicity and membrane affinity.”

Reviewer 2 Report

This review is nicely focusing on the protein intrinsic disorder regions in autophagy factors. The authors nicely summarized most of ATG proteins that contain intrinsic disorder regions and described the membrane interaction mediated by these intrinsic disorder regions. The review manuscript is nicely written and clear for interpretation. There is only one suggestion for the author’s consideration, although it focuses on membrane interaction, could the author add a short paragraph to briefly introduce the role of the intrinsic disorder regions in autophagy related proteins in phase separation? For example, ATG1/ATG13 (Nature. 2020. PMID: 32025038).

Author Response

We thank the reviewer for the suggestion.

This review is nicely focusing on the protein intrinsic disorder regions in autophagy factors. The authors nicely summarized most of ATG proteins that contain intrinsic disorder regions and described the membrane interaction mediated by these intrinsic disorder regions. The review manuscript is nicely written and clear for interpretation. There is only one suggestion for the author’s consideration, although it focuses on membrane interaction, could the author add a short paragraph to briefly introduce the role of the intrinsic disorder regions in autophagy related proteins in phase separation? For example, ATG1/ATG13 (Nature. 2020. PMID: 32025038).

RESPONSE. Thank you for pointing this out. We added a short paragraph on page 8, and the Nature article is now include in the list of the references as ref. # [37].

Reviewer 3 Report

The manuscript entitled: "Prevalence of the protein intrinsic disorder-based interactions of the autophagy machinery with membranes", by Popelka and Uversky, reviews the molecular mechanisms adopted by intrinsically disordered proteins and/or protein regions to modulate membrane properties in the autophagy pathway. The nature of the chemical interactions involved is presented, together with examples of induced folding upon membrane binding, effect of post-translational modifications and discussion on the open questions in the field.

The theme is very interesting and developed in a deep and rigorous way. My main concern about the actual version of the manuscript is related to the extensive employment (as highlighted by figures 2-11) of advanced computational approaches, such as AlphaFold modeling. These novel methods are revolutionizing protein structural predictions, and in my opinion their description deserves some paragraphs in the text. What are the challenges posed by IDPs to structural modeling? How artificial intelligence programs deal with them? How is AlphaFold employed in this work?

Minor points:

1) the Title is a bit convoluted. I suggest to find a shorter and more appealing one.

2) In most of the figures the panels of disorder predictors are very unclear. I suggest removing the boxes with the legend (already described in the captions), increase the size of numbers and axis titles, and increase picture resolution. 

Author Response

We thank the reviewer for the appreciation, suggestion and minor points.

The theme is very interesting and developed in a deep and rigorous way. My main concern about the actual version of the manuscript is related to the extensive employment (as highlighted by figures 2-11) of advanced computational approaches, such as AlphaFold modeling. These novel methods are revolutionizing protein structural predictions, and in my opinion their description deserves some paragraphs in the text. What are the challenges posed by IDPs to structural modeling? How artificial intelligence programs deal with them? How is AlphaFold employed in this work?

RESPONSE. Thank you for pointing this out. The corresponding discussion is added to the revised manuscript (see legend to Figure 2). In our view, a discussion of the challenges posed by IDPs to structural modeling and a description of how artificial intelligence programs deal with these challenges are outside the scope of this review.

Minor points:

1) the Title is a bit convoluted. I suggest to find a shorter and more appealing one.

RESPONSE. Thank you for pointing this out. We changed the title to read: “Theatre in the self-cleaning cell: Intrinsically disordered proteins or protein regions acting with membranes in autophagy.”

2) In most of the figures the panels of disorder predictors are very unclear. I suggest removing the boxes with the legend (already described in the captions), increase the size of numbers and axis titles, and increase picture resolution. 

RESPONSE. We appreciate this point, it improved our figures. Figures were modified as suggested. We also added a new table which provides information on the localization of the IDRs in all the proteins discussed in this study. In our view, this information will help in comprehension of the reported data.

Round 2

Reviewer 3 Report

According to this referee the revised version of the manuscript is OK for publication.